# Infertility screening in unmarried men: A scoping review protocol

Sanam Borji-Navan[1], Nasser Mogharabian[2*]

1 Student Research Committee, School of Nursing and Midwifery, Shahroud University of Medical Sciences, Shahroud, Iran, 2 Assistant professor of andrology and urology, Sexual Health and Fertility Research Center, Shahroud University of Medical Sciences, Shahroud, Iran

* dr.mogharabian@gmail.com

## Abstract

### Introduction

Infertility is often viewed as a couple-centric issue; however, male infertility significantly contributes to reproductive challenges. While infertility screening is increasingly discussed in partnered contexts, limited evidence exists specifically regarding infertility screening for unmarried men. This protocol outlines a scoping review that aims to explore the scope and nature of evidence related to infertility screening in unmarried men.

### Methods

This scoping review will adhere to a comprehensive 14-step methodological framework, incorporating the established Arksey and O'Malley methodology, enhanced by Tricco and Peters. It will follow the PRISMA-ScR guidelines. A combination of thesaurus and free-text search methods will be employed, connecting keywords within each concept. Databases including Web of Science (ISI), PubMed, Scopus, and search engines like Google Scholar, will be searched. To ensure rigor, the selection process will be conducted using the established PCCT framework (population, concept, context, and study type). This review will consider quantitative, qualitative, and mixed-methods studies, as well as grey literature from any geographical location and setting, focusing on infertility screening concepts for unmarried men. The extracted data will be synthesized and presented through diagrams and tables, accompanied by a narrative summary.

### Discussion

This scoping review will summarize the evidence on infertility screening in unmarried men, highlighting diverse types of screening, potential benefits and harms, and knowledge gaps. It aims to guide future research and improve the understanding of the reproductive health needs of unmarried men worldwide.

**Data availability statement:** No datasets were generated or analysed during the current study. All relevant data from this study will be made available upon study completion.

**Funding:** The author(s) received no specific funding for this work.

**Competing interests:** The authors have declared that no competing interests exist.

## Introduction

Infertility, a global health concern affecting millions of people worldwide, is often seen as a couple-centric issue [1]. However, male factors significantly contribute to infertility challenges in nearly half of all cases, underscoring the critical importance of addressing male reproductive health [2].

While discussions about fertility screening and assessment are becoming increasingly common, particularly among couples planning to conceive, the evidence landscape regarding infertility screening for unmarried men remains unexplored. This knowledge gap is not merely a demographic oversight but represents a fundamental clinical and public health distinction [3]. The entire infrastructure of fertility assessment is overwhelmingly built on a reactive, couple-centric model, where screening for male factors is initiated almost exclusively after a couple has failed to conceive (i.e., couple's infertility). This reactive model is, by definition, inaccessible and irrelevant to unmarried men [4,5]. For this population, screening must be proactive or opportunistic, which shifts the entire paradigm from diagnosing a present, shared problem to assessing future, individual reproductive potential [5,6]. This fundamental shift creates a cascade of unique barriers and clinical questions that do not apply in the couple-based setting, including the absence of a clear trigger for seeking care, low awareness of fertility as an individual health metric, different counseling needs for a hypothetical future, and a near-total lack of clinical guidelines for this invisible population within the current health system [5,7–9].

Infertility screening presents notable advantages when conducted with discretion and appropriate medical oversight [10]. Early detection of reproductive health anomalies, encompassing hormonal imbalances, genetic predispositions, and structural irregularities, facilitates timely interventions, thereby enhancing prospective reproductive outcomes [11–13].

Furthermore, such screenings foster heightened awareness and educational opportunities regarding reproductive health, empowering individuals to make informed lifestyle choices that mitigate potential adverse impacts on fertility [14]. The psychological benefits, including anxiety reduction and reassurance, are significant, particularly for those harboring concerns about future fecundity [15]. Preventive measures, such as addressing sexually transmitted infections [16], can be initiated, and fertility preservation options can be explored [17]. On a public health scale, early screening contributes to improved reproductive health outcomes, diminishing the prevalence of infertility-related complications in adulthood [18].

Infertility screening in unmarried men involves a multifaceted diagnostic protocol encompassing medical history, physical examination, semen analysis [18], hormonal and genetic testing, specialized sperm assessments, imaging, and lifestyle evaluations [19,20]. Preventing male infertility necessitates a multidimensional approach involving lifestyle modifications [21,22], environmental awareness, and proactive medical management [18].

Societal shifts, such as increased age at marriage, a desire for later-life parenthood, and a growing awareness of individual reproductive health, underscore the

significance of infertility screening in unmarried men [23]. Furthermore, cultural and social perspectives influence unmarried men's decisions regarding infertility screening [24,25].

Understanding the current evidence base concerning infertility screening in unmarried men is crucial for identifying the extent and nature of existing research, pinpointing knowledge gaps, and informing future research agendas, clinical practice, and public health initiatives aimed at promoting men's reproductive health throughout their lives. If issues are identified through the screening process, significant steps, such as those mentioned (lifestyle modifications, medical interventions, etc.) [18,21], can be taken. Conversely, the absence of such preventive measures may lead to more severe and complex reproductive health consequences in the future [18]. This scoping review aims to address this critical gap by mapping the available literature on infertility screening in unmarried men, exploring the range of screening approaches, populations studied, and contexts examined, as well as identifying key themes and areas that require further attention.

A preliminary search of PROSPERO, Cochrane Library, Google Scholar, and JBI Evidence Synthesis was conducted. To the best of our knowledge, no existing or ongoing systematic reviews or scoping reviews on this topic were found. The objective of this scoping review is to evaluate the current literature on infertility screening in unmarried men.

## Objectives

### Primary outcomes.

1. Identify and map existing guidelines for male infertility screening and assess their applicability and recommendations specifically regarding unmarried men.

2. Map the range and types of infertility screening methods currently used/can be used for unmarried men.

### Secondary outcomes.

1. To explore the perspectives and attitudes towards infertility screening.

2. To synthesize the potential benefits (pros) and harms/drawbacks (cons) of infertility screening in unmarried men.

3. To identify knowledge gaps and areas for future research related to infertility screening in unmarried men and policy implications.

## Review question

What is the scope and nature of the evidence regarding infertility screening in unmarried men?

**Inclusion and exclusion criteria.** To ensure the rigor and relevance of the included studies, the selection process will be conducted using the established PCCT framework. A detailed explanation of the PCCT criteria applied in this review is presented in Table 1.

We exclude unmarried men who were receiving infertility treatment. This is because our study focuses on the screening stage, not the subsequent treatment stage. These populations have distinct clinical characteristics and informational needs. As this is a scoping review, the objective is to map the full range of evidence, not to quantitatively synthesize it. Therefore, contextual data will be charted descriptively to identify where research is concentrated and where gaps exist.

The inclusion and exclusion criteria may be refined iteratively during the initial search and screening phases as the reviewers gain a better understanding of the available literature and the scope of the evidence. Any modifications to this protocol will be documented and reported in the final scoping review. This research will utilize machine translation to analyze all relevant scholarly articles, irrespective of language, thereby ensuring a thorough and impartial review. The

**Table 1. PCCT framework.**

| | Property | Inclusion criteria | Exclusion criteria |
|---|---|---|---|
| P | Participants | Unmarried men in any ages. | Unmarried men who were receiving infertility treatment.<br>Men with a history of previous fertility or attempts at conception. |
| C | Concept | Infertility Screening (any assessment, test, protocol, or practice used to identify the risk of, or early indicators of, infertility in unmarried men who are asymptomatic, not yet diagnosed, and not actively seeking fertility solutions.) | Populations undergoing treatment (any medical intervention, surgical procedure, or assisted reproductive technology (ART) provided to an individual after a formal diagnosis of infertility has been made). |
| C | Context | Any geographical location and setting (e.g., specialized fertility clinics, primary care, community settings). | – |
| T | Types of sources | Quantitative and Qualitative studies, Mixed-methods studies, Grey literature (reports, guidelines, policy documents, conference abstracts and proceedings (if they provide sufficient detail), dissertations). | Editorials, letters, commentaries without original data, Reviews, Duplicates. |

literature search will span from July 1, 1990, to March 30, 2025, capturing the most up-to-date evidence and aligning with the updating policies of the consulted databases.

## Methods and analysis

The protocol for this review follows the guidelines set by the Preferred Reporting Items for Systematic Reviews and Meta-Analyses Protocols (PRISMA-P) (S1 Checklist) [26]. The completed scoping review will be reported in accordance with the PRISMA-ScR guideline [27]. This scoping review will follow a robust 14-step framework that combines the established Arksey and O'Malley [28] methodology with significant enhancements suggested by Tricco and Peters [29,30]. This approach ensures thorough and transparent investigation. The comprehensive framework will guide the review process through stages, including:

1. Develop the research protocol.
2. Define the research question and objectives.
3. Establish inclusion/exclusion criteria.
4. Search relevant resources.
5. Evaluate reference lists.
6. Search grey literature.
7. Screen studies/articles.
8. Study selection.
9. Design a data charting template.
10. Execute data charting.
11. Present results in tables and figures.
12. Provide a flowchart.
13. Identify research implications.
14. Identify practical implications.

**Key Definitions and Scope**

○ **Population:** Our focus on unmarried men is driven by a fundamental clinical and public health distinction. Current fertility care is overwhelmingly based on a reactive, couple-centric model, where male screening is initiated after a couple fails to conceive. This model is, by definition, inaccessible to unmarried men. Our review focuses on the necessary shift to a proactive or opportunistic model for this population. This involves assessing future, individual reproductive potential rather than diagnosing a present, shared problem, which creates unique barriers, counseling needs, and clinical questions that justify this specific focus. To ensure internal validity and focus on true screening, men with a history of proven fertility or previous attempts at conception will be excluded.

○ Concept: For the purpose of this review, infertility screening refers to any assessment, test, protocol, or practice used to identify the risk of, or early indicators of, infertility in unmarried men who are asymptomatic, not yet diagnosed, and not actively seeking fertility solutions. Treatment is defined as any medical, surgical, or ART intervention provided after a formal diagnosis of infertility has been made. Therefore, studies focusing exclusively on treatment of populations will be excluded.

○ Context: This review will include studies from any geographical location or setting (e.g., primary care, specialized clinics, community settings). As this is a scoping review, the objective is to descriptively map the full range of evidence, not to quantitatively synthesize it. Contextual data (such as country, income level, and setting) will be charted to identify where research is concentrated and where significant gaps exist.

**Search methods and sources (Search strategy)**

The reporting in this section will comply with the PRISMA-S checklist guidelines [31]. A thorough search strategy will be developed to ensure the retrieval of all pertinent literature. This will involve systematically organizing search terms into key concepts and identifying relevant keywords that align with the study objectives and inclusion criteria. This meticulous methodology aims to enhance the comprehensiveness of literature review while reducing the likelihood of omitting significant studies.

This study will adopt a multifaceted approach to identify relevant keywords and phrases. Established thesauri, such as MeSH, EMTREE, and the ERIC Thesaurus, will be used alongside free-text methods, including the review of related articles and specialized books. Expert opinions will also be sought to capture emerging terminologies, ensuring a robust set of keywords for the study.

Search strategies will be carefully tailored for each database and search engine to optimize the retrieval of relevant results. Keywords representing similar concepts will be combined using "OR" to broaden the search scope, while distinct concepts will be linked with "AND" to ensure precise results. S2 File contains an initial search strategy for PubMed. This strategy is a starting point and can be improved. The Polyglot Search Translator will be used to translate searches between different databases [32]. This systematic approach aims to capture a wide range of relevant literature. A detailed log of the search terms and strategies used for each database will be included in Supplementary File 1, ensuring transparency and reproducibility.

To achieve a thorough overview of the relevant literature, a multifaceted search strategy will be implemented. This strategy, which will be executed by researcher SB, will encompass a variety of prominent research databases, including Web of Science (ISI), PubMed, and Scopus. In addition to these traditional databases, search engines such as Google Scholar will be utilized to broaden the scope and identify potentially relevant studies from a wider array of sources. Articles will be subjected to bibliometric analysis through forward and backward citation tracking. The relevance and context of these articles will be assessed independently by two reviewers.

## Study records

**Data management.** All the records found will be uploaded to Rayyan [33], where duplicate entries will be identified and removed by S.B.. Following an initial review of the titles and abstracts by two independent reviewers, full-text articles will be uploaded for a comprehensive evaluation by two independent reviewers, while any unsuitable records will be managed separately. This approach will ensure efficient organization and thorough analysis of literature.

**Selection process.** The study selection process will be conducted in two distinct phases. First, titles and abstracts will be screened independently by two reviewers, and studies will be categorized as included, probably included, or excluded. Both included and probably included studies will then be retrieved for a full-text eligibility assessment. Up to three attempts will be made to contact the corresponding authors of potentially eligible studies if the full text is not available. Studies that remain irretrievable will not be excluded; instead, their total number will be reported in the PRISMA flow diagram and addressed as a study limitation. Then, the full texts of the remaining articles will be independently assessed by two reviewers based on pre-defined inclusion and exclusion criteria. Any discrepancies between the reviewers will be resolved through consensus, or, if necessary, by consulting a third-party arbitrator.

A PRISMA flowchart (Fig 1) will be utilized to transparently document the selection process of articles, providing detailed justifications for inclusion or exclusion. The authors of the studies will be contacted as necessary to obtain any missing data.

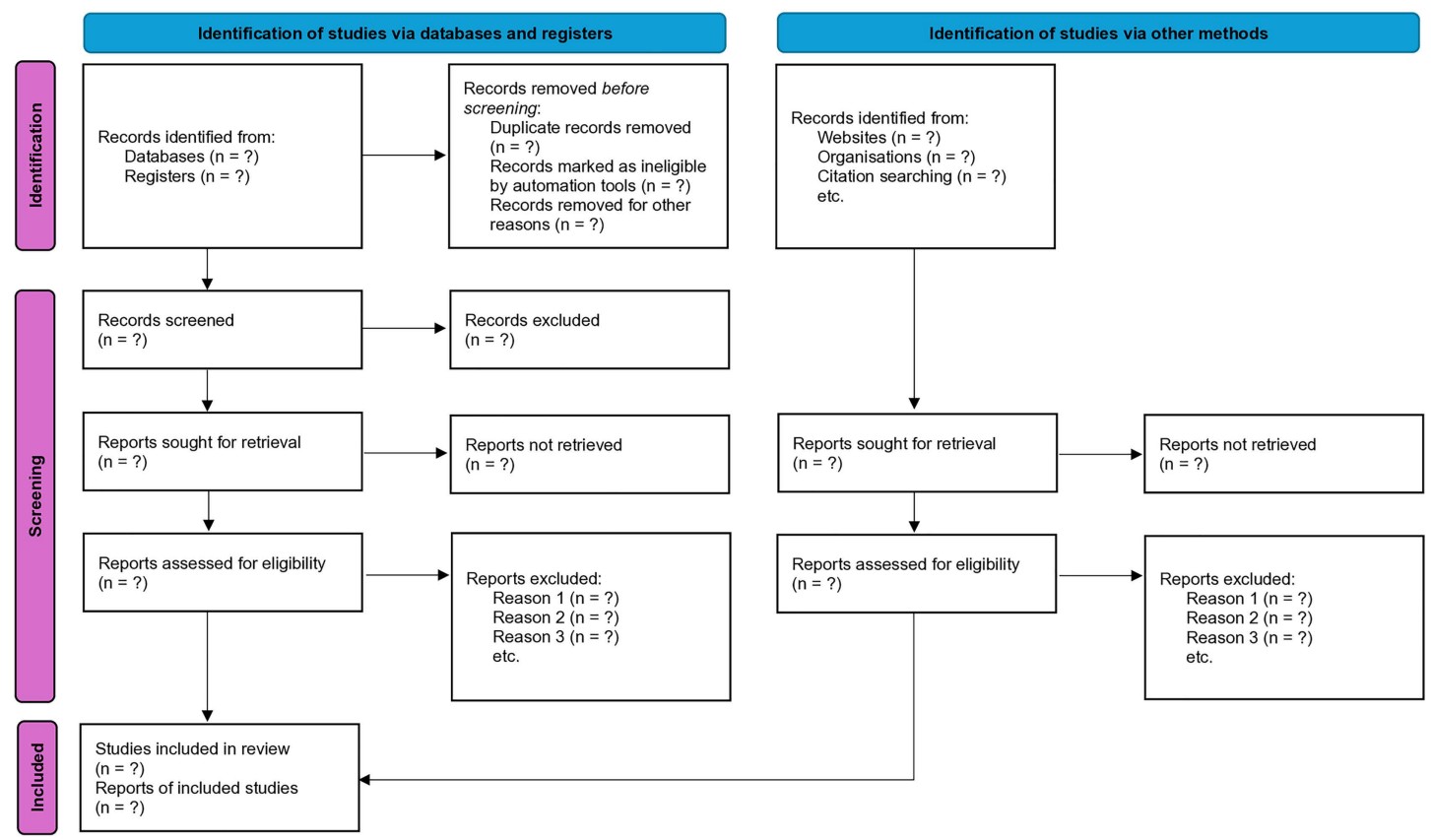

**Fig 1. PRISMA flowchart.**

### Data collection and analysis

Two independent extractors will meticulously review the full text of each study meeting the inclusion criteria. A standardized data charting form, rigorously developed and piloted by the research team, will guide the systematic extraction of relevant data. This form is designed to align with the PCCT framework and the objectives of this scoping review. The data charting form will capture key study characteristics, broadly categorized as follows:

○ **Study characteristics:** (authors, year of publication, study design, country of study, language of publication).

○ **Participant Characteristics:** (Description of the population, e.g., age range, specific subgroups if mentioned, recruitment methods, sample size, relevant contextual details about the population studied).

○ **Concept: Infertility Screening Details:** (Type of infertility screening method(s) discussed or studied, reasons/motivations for screening explored (if any), barriers and facilitators to screening identified (if any), ethical and social considerations mentioned (if any), policy implications or recommendations (if any), reported awareness or accessibility aspects).

○ **Contextual Details:** (Setting where the study was conducted—e.g., clinical, community, public health program, research setting; relevant geographical or cultural context influencing the findings).

○ **Study Findings/Key Messages:** Summary of the main findings related to infertility screening in unmarried men; identification of knowledge gaps or future research areas suggested by the authors.

The data extraction process will be iterative. The data charting form will be reviewed and refined after initial extractions from a subset of studies to ensure all relevant variables are captured comprehensively and accurately for the review's objectives. Discrepancies in data extraction between the two extractors will be resolved through discussion and, if necessary, consultation with a third reviewer. The extracted data will then be synthesized and presented using tables and diagrams to map the evidence, accompanied by a narrative summary highlighting key themes, gaps, and the overall scope of research in this area.

S.B. will be contacted as needed to obtain any missing information considered essential for comprehensive data synthesis, thereby ensuring the completeness of the extracted data. A rigorous quality assurance process will be implemented to address potential discrepancies between the two independent extractors. This process will involve consultation with a third researcher to achieve consensus and ensure the accuracy of the extracted data. Subsequently, the finalized data will be systematically categorized and organized to facilitate efficient and effective analysis.

### Critical Appraisal

Consistent with the primary objective of a scoping review, which is to comprehensively map the extent, range, and nature of evidence on a topic rather than to synthesize findings based on study quality, a formal quality appraisal of the included studies will not be performed [27].

### Data synthesis and analysis

Data synthesis will be conducted using an inductive thematic analysis. The synthesized findings will then be presented through diagrams and tables, accompanied by a narrative summary to provide context and further elaboration in accordance with the review's objectives.

### Discussion

This scoping review protocol outlines a rigorous methodology to map the existing evidence regarding infertility screening in unmarried men. By systematically exploring a broad range of sources and study types, this review is anticipated to make significant contributions to the field of men's reproductive health.

## Strengths and limitations

This study has several strengths. It utilizes a comprehensive search strategy across multiple databases and grey literature, ensuring a wide coverage of available evidence. The use of the methodological framework by Arksey and O'Malley [28], enhanced by Tricco and Peters [29,30], along with dual independent screening and data extraction, ensures the rigor and reproducibility of the review. However, limitations may exist. However, limitations may exist. Although translation tools will be utilized to minimize language bias, potential limitations regarding the accuracy of translation or the loss of cultural and linguistic nuances in non-English and non-Persian studies may still exist. Additionally, given the novelty of the topic, there may be a scarcity of studies explicitly focusing on unmarried men, requiring careful interpretation of data from broader male populations.

## Implications for policy and practice

The findings of this scoping review are expected to have implications for policy and practice. By summarizing policy recommendations and assessing reported awareness and accessibility of screening services, the review can inform the development of more inclusive and equitable reproductive health policies and programs that cater to the diverse needs of men, regardless of their marital status. Specifically, moving from a reactive model to a proactive screening approach requires evidence-based guidelines. This review will provide the necessary foundational knowledge to support this shift, advocating for programs that cater to the diverse needs of men, regardless of their marital status.

## Future research

Critically, this scoping review will identify key knowledge gaps and areas for future research. By systematically mapping the existing evidence, the review will pinpoint areas where research is lacking, highlighting priorities for future investigations. This will be instrumental in guiding researchers and funding bodies to focus efforts on addressing the most pressing unanswered questions related to infertility screening and the reproductive health of unmarried men.

In conclusion, this scoping review protocol provides a robust framework for examining the evidence landscape surrounding infertility screening in unmarried men. The anticipated outputs of this review will be invaluable in informing future research, guiding ethical considerations, and contributing to improved reproductive health services and outcomes for unmarried men globally.

## Ethics and dissemination

Any amendments will be documented in the final publication. Due to the anticipated breadth and complexity of the data, the results of this scoping review may be disseminated in multiple publications, each focusing on specific aspects of the research questions and objectives.

## Supporting information

**S1 Checklist. PRISMA-P 2015 checklist.**
(DOCX)

**S2 File. Search strategy.**
(DOCX)

## Acknowledgments

The authors acknowledge the use of an AI language model for assistance with translating from Persian to English, writing, and editing this manuscript. The authors maintained full oversight and responsibility for the final content.

## Author contributions

**Conceptualization:** Sanam Borji-Navan, Nasser Mogharabian.

**Methodology:** Sanam Borji-Navan, Nasser Mogharabian.

**Project administration:** Sanam Borji-Navan.

**Supervision:** Nasser Mogharabian.

**Validation:** Nasser Mogharabian.

**Writing – original draft:** Sanam Borji-Navan.

**Writing – review & editing:** Sanam Borji-Navan, Nasser Mogharabian.

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
