## [Decision Letter · Decision Letter 0]

5 May 2025

Dear Dr. Mogharabian,

Thank you for submitting your manuscript to PLOS ONE. After careful consideration, we feel that it has merit but does not fully meet PLOS ONE’s publication criteria as it currently stands. Therefore, we invite you to submit a revised version of the manuscript that addresses the points raised during the review process.

Key Areas for Improvement:

Work on the total organization of the paper (Reviewer #1 and #2).

In summary, I encourage you to address all the reviewers' comments and make the necessary revisions. I look forward to reviewing your revised manuscript.

We look forward to receiving your revised manuscript.

Kind regards,

Godwin Banafo Akrong, Ph.D.

Academic Editor

PLOS ONE

Journal Requirements:

2. In the online submission form, you indicated that “Data will be available upon reasonable request to the corresponding author.”

Reviewers' comments:

Reviewer's Responses to Questions

**Comments to the Author**

1. Does the manuscript provide a valid rationale for the proposed study, with clearly identified and justified research questions?

Reviewer #1: Yes

Reviewer #2: Yes

2. Is the protocol technically sound and planned in a manner that will lead to a meaningful outcome and allow testing the stated hypotheses?

Reviewer #1: Yes

Reviewer #2: Partly

3. Is the methodology feasible and described in sufficient detail to allow the work to be replicable?

Reviewer #1: Yes

Reviewer #2: No

4. Have the authors described where all data underlying the findings will be made available when the study is complete?

Reviewer #1: Yes

Reviewer #2: No

5. Is the manuscript presented in an intelligible fashion and written in standard English?

Reviewer #1: Yes

Reviewer #2: Yes

You may also provide optional suggestions and comments to authors that they might find helpful in planning their study.

Reviewer #1: Thank you for submitting this protocol. This is an interesting subject in our daily fertility care.

I have few comments to address.

1. The PCCT framework table formatting is confusing in its current form. Please revise for clarity

2. It is not clear what kind of translation (as mentioned in the acknowledgement) has been conducted using AI.

3. While it is common practice in a scoping review to skip formal quality appraisal, a brief justification would strengthen the rationale. We have faced this issue before when top tier journals demanded that we assess the quality of included studies in a scoping review.

4. A final language check would improve readability of the manuscript.

Reviewer #2: The article is written like a proposal. It is futuristic and reads more like a scoping review guideline for the authors, than like a completed research protocol. It is clear on the existing gap in literature about infertility screening in unmarried men and captures what the authors research intents are, including the protocol they plan to adopt. As a guideline for embarking on the scoping review, it offers sufficient detail on the processes and procedures it plans to follow. However, it is lacking in detail as to whether these activities have actually been undertaken, how they were undertaken, the processes for administering or applying the PRISMA-ScR protocol, what the outcomes are and the implications for future research in that field.

As it stands, the scoping review is yet to be conducted.

Other Observations:

In line 65, mention is made of polycystic ovary syndrome, this is not a factor in male infertility.

In line 173, 'we've' should be written in full as 'we have'.

**Do you want your identity to be public for this peer review?** For information about this choice, including consent withdrawal, please see our Privacy Policy

Reviewer #1: No

Reviewer #2: No

---

## [Author Response · Author response to Decision Letter 1]

9 May 2025

Dear Dr. Godwin Banafo Akrong, 2025/05/09

Academic Editor of PLOS ONE

Thank you for your email and the opportunity to revise our manuscript. We appreciate the time and insights of the reviewers and believe that their comments have helped us to significantly improve the manuscript. We have carefully considered all of the comments and have revised the manuscript accordingly.

Below, we provide a point-by-point response to each of the editor and reviewers' comments.

Journal Requirements:

Comment 1. Please ensure that your manuscript meets PLOS ONE's style requirements, including those for file naming. The PLOS ONE style templates can be found at https://journals.plos.org/plosone/s/file?id=wjVg/PLOSOne_formatting_sample_main_body.pdf and https://journals.plos.org/plosone/s/file?id=ba62/PLOSOne_formatting_sample_title_authors_affiliations.pdf.

Response: Thank you for this important point. We have carefully reviewed the PLOS ONE style guidelines and made the necessary revisions to ensure compliance. As these revisions primarily involved formatting adjustments, such as file naming and layout, we have not highlighted them. However, we confirm that all changes have been made according to the journal's requirements.

Comment 2. In the online submission form, you indicated that “Data will be available upon reasonable request to the corresponding author.”

Response: No datasets were generated or analysed during the current study. All relevant data from this study will be made available upon study completion. The sentence was corrected.

Comment 3. Please include a separate caption for each figure in your manuscript.

Response: We revise the manuscript to include a separate caption for each figure, as requested. (Page 8, Line 183)

Reviewer #1:

Thank you for submitting this protocol. This is an interesting subject in our daily fertility care.

I have few comments to address.

Comment 1. The PCCT framework table formatting is confusing in its current form. Please revise for clarity.

Response: We have completely reformatted the PCCT table. (Page 5, Line 106)

Comment 2. It is not clear what kind of translation (as mentioned in the acknowledgement) has been conducted using AI.

Response: Thank you for raising this point. We revised this section. (Page 12, Lines 255-256)

Comment 3. While it is common practice in a scoping review to skip formal quality appraisal, a brief justification would strengthen the rationale. We have faced this issue before when top tier journals demanded that we assess the quality of included studies in a scoping review.

Response: Thank you for your insightful comment regarding the justification for not performing a formal quality appraisal. We appreciate the suggestion to strengthen our rationale. we have revised this section. (Page 11, Lines 222-224)

Comment 4. A final language check would improve readability of the manuscript.

Response: Thank you for your suggestion. We perform a language check to improve clarity.

Reviewer #2:

The article is written like a proposal. It is futuristic and reads more like a scoping review guideline for the authors, than like a completed research protocol. It is clear on the existing gap in literature about infertility screening in unmarried men and captures what the authors research intents are, including the protocol they plan to adopt. As a guideline for embarking on the scoping review, it offers sufficient detail on the processes and procedures it plans to follow. However, it is lacking in detail as to whether these activities have actually been undertaken, how they were undertaken, the processes for administering or applying the PRISMA-ScR protocol, what the outcomes are and the implications for future research in that field.

As it stands, the scoping review is yet to be conducted.

Response:. Thank you for your precise feedback. As our title "A Scoping Review Protocol" indicates, this manuscript is indeed a protocol, not a completed study. Therefore, the details on execution and outcomes, which you correctly noted are absent, are not yet available as the review is yet to be conducted. Our aim was to transparently present a detailed research plan prior to its execution.

Other Observations:

Comment 1. In line 65, mention is made of polycystic ovary syndrome, this is not a factor in male infertility.

Response:. Thank you for your careful review and for identifying this error. You are absolutely correct; Polycystic Ovary Syndrome (PCOS) is related to female infertility and is not a factor in male infertility, which is the topic of our study. We apologize for this oversight. We have removed the mention of Polycystic Ovary Syndrome in the revised manuscript to ensure the text accurately reflects factors relevant to male infertility.

Comment 2. In line 173, 'we've' should be written in full as 'we have'.

Response: Thank you for this comment. We agree that formal academic writing standards should be maintained. As suggested, we have changed 'we've' to 'we have' in the revised manuscript. We have also checked the document.

Response: We add a figure in this manuscript with requirements checking.

---

## [Decision Letter · Decision Letter 1]

25 Aug 2025

Dear Dr. Mogharabian,

Thank you for submitting your manuscript to PLOS ONE. After careful consideration, we feel that it has merit but does not fully meet PLOS ONE’s publication criteria as it currently stands. Therefore, we invite you to submit a revised version of the manuscript that addresses the points raised during the review process.

Alongside addressing all the Reviewers' concerns, please specifically focus on the following:

The methods should be detailed enough to ensure reproducibility and prevent undisclosed flexibility.Include appropriate controls, sample size calculations, and replication to ensure robust and reproducible data.The exploratory aspects of the analysis should be explicitly described.Reference was made to the PRISMA-ScR framework, but the protocol does not fully adhere to its 14-step process.Specify whether data synthesis will be conducted inductively, deductively, or both.The exclusion criteria and data synthesis process require further refinement and clarification.

We look forward to receiving your revised manuscript.

Kind regards,

Godwin Banafo Akrong, Ph.D.

Academic Editor

PLOS ONE

Journal Requirements:

Reviewers' comments:

Reviewer's Responses to Questions

**Comments to the Author**

1. Does the manuscript provide a valid rationale for the proposed study, with clearly identified and justified research questions?

Reviewer #2: Yes

Reviewer #3: Yes

2. Is the protocol technically sound and planned in a manner that will lead to a meaningful outcome and allow testing the stated hypotheses?

Reviewer #2: Yes

Reviewer #3: Yes

3. Is the methodology feasible and described in sufficient detail to allow the work to be replicable?

Reviewer #2: Yes

Reviewer #3: Yes

4. Have the authors described where all data underlying the findings will be made available when the study is complete?

Reviewer #2: Yes

Reviewer #3: Yes

5. Is the manuscript presented in an intelligible fashion and written in standard English?

Reviewer #2: Yes

Reviewer #3: Yes

You may also provide optional suggestions and comments to authors that they might find helpful in planning their study.

Reviewer #2: The authors indicate that their submission is a scoping review protocol designed to transparently present a detailed research plan prior to its execution, not the actual completed review. The article offers comprehensive insights into what the authors intend to do, and evidences an in-depth review of literature on the subject matter.

However, I doubt that the use of the word 'protocol' to describe the article is appropriate, particularly as they allude to the use of the PRISMA-ScR which is a known, widely used scoping review protocol. Consequently, referring to their submission as a protocol may only confuse the intended audience, since this submission actually reads like a research design/methodology that is focused on a specific area of research.

They state in lines 16 and 17 that 'the aim of this scoping review is to understand the scope and nature of evidence related to infertility screening in unmarried men'; and in lines 37 to 39, that 'this scoping review will summarize the evidence on infertility screening in unmarried men, highlighting diverse types of screening, potential benefits and harms/drawbacks of infertility screening, and knowledge gaps.' These statements seem to imply that the article is a scoping review report, yet it only reflects some, not all of the 14-step framework outlined in lines 124 to 137, and so is inconclusive.

On this basis, I advise that the scoping review be conducted and properly reported, such that it effectively captures all the stages of the 14-step framework, and the primary and secondary outcomes outlined in the objective. This will prevent any lack of clarity or confusion on the actual focus of the article.

Thereafter, it can be resubmitted for consideration for publication.

Reviewer #3: Thank you for the opportunity to review this manuscript. The authors planned to review the scope of the literature on screening for male infertility among unmarried men. This is a usual research piece.

Find my few comments below:

Abstract:

1. Structure the abstract as follows: Background/Introduction including the primary goal, Methods including the eligibility criteria, Results (not applicable), Conclusion (not applicable). The protocol should focus on the background, objective and methods.

Methods

1. This is a study protocol, kindly use future tenses not past tense. See lines, 112, 113, 140, 154, 158, 165, 166, 175, 176 and so on. Replace "was, were" with "will" as appropriate.

2. In the exclusion criteria, remove the exclusion of married men. This is because the study included only unmarried men, hence, by definition married men are excluded.

3. Consider including only original articles with primary or may be secondary data. I don't think review articles should be included.

3. Exclusion of articles should be only based on not meeting the exclusion criteria, not necessarily those who the index authors could get the full article from the authors. However, if an article is considered to be included but the authors could not get access to the full manuscript, a consideration could be made to extract information from the abstract or consider these set of studies as a possible limitation for the scoping review.

4. The authors categorized the selected articles as included, probably included and excluded. How will the "probably included" be finally disposed?

5. The process of data synthesis: Is this going to be done inductively or deductively or both?

**Do you want your identity to be public for this peer review?** For information about this choice, including consent withdrawal, please see our Privacy Policy

Reviewer #2: No

Reviewer #3: No

---

## [Author Response · Author response to Decision Letter 2]

29 Aug 2025

Dear Dr. Godwin Banafo Akrong, 2025/08/25

Academic Editor of PLOS ONE

Thank you for your email and the opportunity to revise our manuscript. We appreciate the time and insights of the reviewers and believe that their comments have helped us to significantly improve the manuscript. We have carefully considered all of the comments and have revised the manuscript accordingly.

Below, we provide a point-by-point response to each of the editor and reviewers' comments.

Comment 1. The methods should be detailed enough to ensure reproducibility and prevent undisclosed flexibility.

Response: We thank the reviewer for emphasizing the critical importance of reproducibility and methodological transparency. Guided by this principle, and incorporating the other specific suggestions from the reviewers, we have revised the Methods section to enhance its level of detail and clarity.

Comment 2. Include appropriate controls, sample size calculations, and replication to ensure robust and reproducible data.

Response: We thank the reviewer for their feedback. We wish to clarify that as a protocol for a scoping review, our methodology is based on knowledge synthesis. Therefore, elements of primary experimental studies such as 'control groups' and 'sample size calculations' are not applicable. Robustness is ensured through a detailed and transparent method designed for reproducibility, as is standard for this type of review.

Comment 3. The exploratory aspects of the analysis should be explicitly described.

Response: We thank the reviewer for this valuable suggestion. We have revised the Methods section to explicitly describe the exploratory nature of our analysis. We have clarified that we will use an inductive thematic analysis to allow themes and patterns to emerge directly from the data, which aligns with the exploratory goals of a scoping review.

Comment 4. Reference was made to the PRISMA-ScR framework, but the protocol does not fully adhere to its 14-step process.

Response: We thank the reviewer for highlighting the need for this important clarification. We wish to explain the distinct roles of the two frameworks mentioned in our manuscript. The 14-step process is the methodology we will follow to conduct the review. This protocol adheres to that process by prospectively detailing how each of the 14 steps will be executed. The PRISMA-ScR, on the other hand, is the guideline we will use for reporting the findings in the final, completed manuscript. We believe this distinction resolves the perceived discrepancy. We have also revised the manuscript to state this separation more explicitly to prevent confusion for readers.

Comment 5. Specify whether data synthesis will be conducted inductively, deductively, or both.

Response: We thank the reviewer for this question regarding our synthesis method. Our data synthesis will be primarily inductive. We have clarified in the manuscript that we will use thematic analysis to allow themes and patterns to emerge directly from the data, consistent with the exploratory nature of a scoping review.

Comment 6. The exclusion criteria and data synthesis process require further refinement and clarification.

Response: We thank the reviewer for this summary feedback. In line with the reviewer's specific comments, we have thoroughly revised both the exclusion criteria and the data synthesis process. We have added significant clarification to both sections to ensure they are methodologically robust and transparent, and we believe our revisions fully address the reviewer's concerns.

Journal Requirements:

Comment 1. If the reviewer comments include a recommendation to cite specific previously published works, please review and evaluate these publications to determine whether they are relevant and should be cited. There is no requirement to cite these works unless the editor has indicated otherwise.

Response: We thank the editor for the clear guidance on handling reviewer-suggested citations. We confirm we have followed this principle throughout our revision process.

Reviewer #1:

Comment 1. The authors indicate that their submission is a scoping review protocol designed to transparently present a detailed research plan prior to its execution, not the actual completed review. The article offers comprehensive insights into what the authors intend to do, and evidences an in-depth review of literature on the subject matter.

However, I doubt that the use of the word 'protocol' to describe the article is appropriate, particularly as they allude to the use of the PRISMA-ScR which is a known, widely used scoping review protocol. Consequently, referring to their submission as a protocol may only confuse the intended audience, since this submission actually reads like a research design/methodology that is focused on a specific area of research.

Response: We thank the reviewer for their valuable feedback on our use of the term 'protocol' and for prompting an important clarification. We have used the term 'protocol' as the manuscript outlines our a priori study plan, which aligns with the standard definition in evidence synthesis. The PRISMA-ScR, in contrast, is the reporting guideline we will follow for the completed review, not the protocol for its execution. To ensure this distinction is perfectly clear to all readers, we have identified the sentence that may have caused ambiguity and have revised it in the manuscript. The sentence now explicitly states: "The completed scoping review will be reported in accordance with the PRISMA-ScR guidelines." (Page 2, Line 3/ Page 6, Line 115). By making this revision, we believe any potential confusion has been removed. Therefore, we have retained the term 'protocol' as it most accurately describes the nature of this manuscript. We are grateful to the reviewer for helping us improve the clarity of our text.

Comment 2. They state in lines 16 and 17 that 'the aim of this scoping review is to understand the scope and nature of evidence related to infertility screening in unmarried men'; and in lines 37 to 39, that 'this scoping review will summarize the evidence on infertility screening in unmarried men, highlighting diverse types of screening, potential benefits and harms/drawbacks of infertility screening, and knowledge gaps.' These statements seem to imply that the article is a scoping review report, yet it only reflects some, not all of the 14-step framework outlined in lines 124 to 137, and so is inconclusive.

On this basis, I advise that the scoping review be conducted and properly reported, such that it effectively captures all the stages of the 14-step framework, and the primary and secondary outcomes outlined in the objective. This will prevent any lack of clarity or confusion on the actual focus of the article.

Thereafter, it can be resubmitted for consideration for publication.

Response: We thank the reviewer for their time and detailed feedback. We believe there may have been a misunderstanding regarding the nature of our manuscript, and we appreciate the opportunity to clarify its scope and purpose. This manuscript is a scoping review protocol, not the report of a completed review. Its specific purpose is to outline the detailed methodological plan before the review is conducted. This practice of publishing protocols is standard in evidence synthesis to ensure transparency and methodological rigor. The reviewer correctly observes that our manuscript describes the aims of the future review (lines 16-17 and 37-39) and details a 14-step framework without reporting on the completion of all steps. This is by design. A protocol is intended to present the plan—covering the initial stages of the framework (e.g., defining the question, criteria, and search strategy) and describing how the subsequent stages (e.g., data charting, analysis, and reporting) will be executed. The current manuscript serves precisely this role. Regarding the reviewer's recommendation to conduct the review and resubmit the full report: we agree this is the essential next phase of our research. The completed scoping review will be written up as a separate manuscript for publication once it is finished. However, the explicit goal of this current submission is to have the methodology peer-reviewed and published as a standalone protocol beforehand. To prevent this confusion for future readers, we have carefully reviewed our abstract and introduction to ensure the term "protocol" is stated prominently and the future tense is used consistently.

We hope this clarifies that our manuscript should be evaluated as a study protocol, not a completed review. We believe that when viewed as a research plan, it is comprehensive and conclusive in its own right.

Reviewer #2:

Thank you for the opportunity to review this manuscript. The authors planned to review the scope of the literature on screening for male infertility among unmarried men. This is a usual research piece.

Find my few comments below:

Abstract:

Comment 1. Structure the abstract as follows: Background/Introduction including the primary goal, Methods including the eligibility criteria, Results (not applicable), Conclusion (not applicable). The protocol should focus on the background, objective and methods.

Response: We thank the reviewer for this constructive suggestion. We agree that this structure improves clarity and immediately identifies the manuscript as a protocol. The abstract has been revised accordingly to include the headings: Introduction, Methods, Results (not applicable), Conclusion (not applicable), and Discussion. (Pages 1-2, Lines 14-37)

Methods

Comment 2. This is a study protocol, kindly use future tenses not past tense. See lines, 112, 113, 140, 154, 158, 165, 166, 175, 176 and so on. Replace "was, were" with "will" as appropriate.

Response: We thank the reviewer for this important correction. The reviewer is correct; as a protocol, the manuscript should be in the future tense. We have now revised the entire document and changed all past tenses to the future tense as requested.

Comment 3. In the exclusion criteria, remove the exclusion of married men. This is because the study included only unmarried men, hence, by definition married men are excluded.

Response: We thank the reviewer for this logical correction. The redundant exclusion criterion for "married men" has been removed from the manuscript as suggested. (Page 5, Line 102)

Comment 4. Consider including only original articles with primary or may be secondary data. I don't think review articles should be included.

Response: We thank the reviewer for this valuable methodological suggestion. We agree, and have revised our eligibility criteria to include only original studies (with primary or secondary data) and to explicitly exclude review articles. (Page 5 102, Line)

Comment 5. Exclusion of articles should be only based on not meeting the exclusion criteria, not necessarily those who the index authors could get the full article from the authors. However, if an article is considered to be included but the authors could not get access to the full manuscript, a consideration could be made to extract information from the abstract or consider these set of studies as a possible limitation for the scoping review.

Response: We thank the reviewer for this crucial methodological point. We have clarified in our protocol that articles will not be excluded based on the unavailability of their full text. Instead, the number of any potentially eligible but irretrievable articles will be reported and discussed as a limitation of the study. (Page 8, Lines 170-172)

Comment 6. The authors categorized the selected articles as included, probably included and excluded. How will the "probably included" be finally disposed?

Response: We thank the reviewer for this request for clarification. The "probably included" category is a temporary designation during the initial title/abstract screening. All these articles will undergo a full-text review, after which a definitive 'include' or 'exclude' decision will be made based on our eligibility criteria. (Page 8, Line 169)

Comment 7. The process of data synthesis: Is this going to be done inductively or deductively or both?

Response: We thank the reviewer for this insightful question. Our data synthesis will be primarily inductive. We will use thematic analysis to allow themes and patterns to emerge directly from the charted data, which is consistent with the exploratory nature of a scoping review. We have clarified this in the manuscript. (Page 11, Lines 223-225)

Response: We add a figure in this manuscript with requirements checking.

---

## [Decision Letter · Decision Letter 2]

22 Oct 2025

Dear Dr. Mogharabian,

Thank you for submitting your manuscript to PLOS ONE. After careful consideration, we feel that it has merit but does not fully meet PLOS ONE’s publication criteria as it currently stands. Therefore, we invite you to submit a revised version of the manuscript that addresses the points raised during the review process.

**I encourage you to address all of Reviewer #4's comments and make the necessary revisions.**

We look forward to receiving your revised manuscript.

Kind regards,

Godwin Banafo Akrong, Ph.D.

Academic Editor

PLOS ONE

Journal Requirements:

Reviewers' comments:

Reviewer's Responses to Questions

**Comments to the Author**

1. Does the manuscript provide a valid rationale for the proposed study, with clearly identified and justified research questions?

Reviewer #4: Partly

2. Is the protocol technically sound and planned in a manner that will lead to a meaningful outcome and allow testing the stated hypotheses?

Reviewer #4: Partly

3. Is the methodology feasible and described in sufficient detail to allow the work to be replicable?

Reviewer #4: No

4. Have the authors described where all data underlying the findings will be made available when the study is complete?

Reviewer #4: No

5. Is the manuscript presented in an intelligible fashion and written in standard English?

Reviewer #4: Yes

You may also provide optional suggestions and comments to authors that they might find helpful in planning their study.

Reviewer #4: Comment 1

Abstract: Please remove results (not applicable), Conclusion (not applicable), just don’t include the heading in the abstract. Just write the Introduction, Methods, and go straight to Discussion.

Comment 2

PCC line 102 under participants: Correct grammar in this; Unmarried men with receiving infertility treatment or married women. What does that mean?

Comment 3

Excludes men "receiving infertility treatment" please clarify

Comment 4

Write this in full; Infertility Screening (methods, protocols, practices, …), this is so unclear/ vague, what specific aspects of screening.

It is important to define this term for clarity: infertility screening – in this protocol what are you referring to? Clearly define the terms screening and treatment such that they are distinguished. What methods are you referring to?

Comment 5

Context – This is too vague, any geographical location and setting, with no exclusion seems very broad appears unrealistically. What are your intentions? Limiting to certain healthcare settings?

1. Are the authors truly interested in ALL contexts equally?

2. How will they synthesize findings across vastly different settings?

3. Will contextual variation be explored as part of the analysis?

Comment 6

After reading this protocol again, I still have so many queries:

While the authors mention "evolving societal norms" and "changing relationship patterns" (lines 48-49), the rationale for specifically focusing on unmarried men rather than all men has not been fully justified.

1. Why does marital status matter for screening approaches, protocols, or outcomes?

2. How do screening considerations differ between married and unmarried men beyond the absence of a partner?

3. What unique barriers, facilitators, or clinical considerations apply specifically to unmarried men?

4. What is driving this focus on unmarried men driven? Is it clinical or something else? Please make it clear and understandable

Comment 7

Perhaps the authors should include a Definitions and Scope section that clearly delineates all ambiguous terms. A protocol is supposed to be reproducible, and this can only be achieved if all sections are clear and well defined. This protocol needs additional work to enhance its reproducibility; clarity is crucial.

**Do you want your identity to be public for this peer review?** For information about this choice, including consent withdrawal, please see our Privacy Policy

Reviewer #4: No

---

## [Author Response · Author response to Decision Letter 3]

23 Oct 2025

Dear Dr. Godwin Banafo Akrong, 2025/10/23

Academic Editor of PLOS ONE

Thank you for your email and the opportunity to revise our manuscript. We appreciate the time and insights of the reviewers and believe that their comments have helped us to significantly improve the manuscript. We have carefully considered all of the comments and have revised the manuscript accordingly.

Below, we provide a point-by-point response to each of the editor and reviewers' comments.

Journal Requirements:

Comment 1. If the reviewer comments include a recommendation to cite specific previously published works, please review and evaluate these publications to determine whether they are relevant and should be cited. There is no requirement to cite these works unless the editor has indicated otherwise.

Response: Thank you for this clarification.

Comment 2. Please review your reference list to ensure that it is complete and correct. If you have cited papers that have been retracted, please include the rationale for doing so in the manuscript text, or remove these references and replace them with relevant current references. Any changes to the reference list should be mentioned in the rebuttal letter that accompanies your revised manuscript. If you need to cite a retracted article, indicate the article’s retracted status in the References list and also include a citation and full reference for the retraction notice.

Response: Thank you for the detailed instructions. We have reviewed our entire reference list for accuracy, completeness, and correct formatting. We utilized Zotero to specifically check for any retracted articles, and we confirm that none were found in our references.

Reviewer #4:

Comment 1. Abstract: Please remove results (not applicable), Conclusion (not applicable), just don’t include the heading in the abstract. Just write the Introduction, Methods, and go straight to Discussion.

Response: Thank you for this clarification. We have revised the abstract structure as requested.

Comment 2. PCC line 102 under participants: Correct grammar in this; Unmarried men with receiving infertility treatment or married women. What does that mean?

Response: Thank you for identifying this error. We apologize for the severe confusion. The line you noted (Unmarried men... or married women) mistakenly listed two of our Exclusion Criteria under the 'Participants' section. We have now revised this. (Pages 5/Lines 110)

Comment 3. Excludes men "receiving infertility treatment" please clarify.

Response: Thank you. We excluded men 'receiving infertility treatment' because our study focuses specifically on infertility screening. Men who are already in treatment have already passed the screening and diagnosis phase. They represent a 'treatment population,' not a 'screening population,' and including them would confuse these two distinct stages of care. We have clarified this rationale in the manuscript. (Pages 5-6/Lines 110-115)

Comment 4. Write this in full; Infertility Screening (methods, protocols, practices, …), this is so unclear/ vague, what specific aspects of screening.

It is important to define this term for clarity: infertility screening – in this protocol what are you referring to? Clearly define the terms screening and treatment such that they are distinguished. What methods are you referring to?

Response: Thank you for this crucial feedback. You are absolutely correct that our initial description 'Infertility Screening (methods, protocols, practices, …)' was unacceptably vague and failed to provide necessary clarity. We sincerely apologize for this major omission. To address this, we have substantially revised the manuscript to include precise operational definitions. (Pages 5-6/Lines 110-115)

Comment 5. Context – This is too vague, any geographical location and setting, with no exclusion seems very broad appears unrealistically. What are your intentions? Limiting to certain healthcare settings?

1. Are the authors truly interested in ALL contexts equally?

2. How will they synthesize findings across vastly different settings?

3. Will contextual variation be explored as part of the analysis?

Response: Thank you for this valid point. We acknowledge that 'any context' appears unrealistically broad. As this is a scoping review, this breadth is intentional. Our primary goal is not to synthesize results, but to map the existing evidence and identify gaps. A central part of our analysis is to extract 'Context' (e.g., country, setting) and descriptively analyze this variation. We have revised the 'Context' section in the manuscript to clarify that our aim is to map and descriptively analyze this contextual variation, not to conduct a quantitative synthesis, which would be unrealistic. (Pages 5-6/Lines 110-115)

Comment 6. After reading this protocol again, I still have so many queries:

While the authors mention "evolving societal norms" and "changing relationship patterns" (lines 48-49), the rationale for specifically focusing on unmarried men rather than all men has not been fully justified.

1. Why does marital status matter for screening approaches, protocols, or outcomes?

2. How do screening considerations differ between married and unmarried men beyond the absence of a partner?

3. What unique barriers, facilitators, or clinical considerations apply specifically to unmarried men?

4. What is driving this focus on unmarried men driven? Is it clinical or something else? Please make it clear and understandable.

Response: Thank you for this critical and insightful feedback. You are absolutely correct that our initial rationale was weak and failed to adequately justify the central premise of our study. We sincerely apologize for this major lack of clarity and have extensively revised the Introduction to make this justification clear and explicit. (Pages 2-3/Lines 45-56)

Comment 7. Perhaps the authors should include a Definitions and Scope section that clearly delineates all ambiguous terms. A protocol is supposed to be reproducible, and this can only be achieved if all sections are clear and well defined. This protocol needs additional work to enhance its reproducibility; clarity is crucial.

Response: Thank you for this excellent and highly constructive suggestion. You are absolutely correct that the protocol's reproducibility was weakened by several ambiguous terms, and we sincerely apologize for this. We have fully adopted your recommendation to improve clarity. We have now added a new dedicated subsection in our Methods section titled Key Definitions and Scope. (Pages 7-8/Lines 145-164)

---

## [Decision Letter · Decision Letter 3]

17 Nov 2025

Dear Dr. Mogharabian,

Thank you for submitting your manuscript to PLOS ONE. After careful consideration, we feel that it has merit but does not fully meet PLOS ONE’s publication criteria as it currently stands. Therefore, we invite you to submit a revised version of the manuscript that addresses the points raised during the review process.

We look forward to receiving your revised manuscript.

Kind regards,

Godwin Banafo Akrong, Ph.D.

Academic Editor

PLOS ONE

Journal Requirements:

Reviewers' comments:

Reviewer's Responses to Questions

**Comments to the Author**

1. Does the manuscript provide a valid rationale for the proposed study, with clearly identified and justified research questions?

Reviewer #3: Yes

Reviewer #4: Yes

2. Is the protocol technically sound and planned in a manner that will lead to a meaningful outcome and allow testing the stated hypotheses?

Reviewer #3: Yes

Reviewer #4: Yes

3. Is the methodology feasible and described in sufficient detail to allow the work to be replicable?

Reviewer #3: Yes

Reviewer #4: Yes

4. Have the authors described where all data underlying the findings will be made available when the study is complete?

Reviewer #3: Yes

Reviewer #4: Yes

5. Is the manuscript presented in an intelligible fashion and written in standard English?

Reviewer #3: Yes

Reviewer #4: Yes

You may also provide optional suggestions and comments to authors that they might find helpful in planning their study.

Reviewer #3: Thank you for the opportunity to review the revised manuscript. The authors have responded satisfactorily to the initial comments. However, a few inconsistencies as shown below:

1. In the abstract section, review articles were included while in the method section (i.e., eligibility), reviews were excluded. I suggest you exclude review articles from the scoping review. If they are previous reviews on the subject, you will need to justify why the current review.

2. Kindly expunge unmarried women from the exclusion criteria. The review focuses on unmarried man. There is no need excluding the unmarried women because by the including only unmarried men, the unmarried women are already excluded.

3. How many members of the team will be involved in reviewing the selection in Raygan?

Reviewer #4: Although the response to my Comment 4 was not entirely satisfactory, I noted that the authors addressed this point in the Key Definitions and Scope section. Overall, the authors have adequately addressed my queries, and I believe the manuscript provides valuable knowledge and is suitable for publication.

**Do you want your identity to be public for this peer review?** For information about this choice, including consent withdrawal, please see our Privacy Policy

Reviewer #3: No

Reviewer #4: No

---

## [Author Response · Author response to Decision Letter 4]

17 Nov 2025

Dear Dr. Godwin Banafo Akrong, 2025/11/17

Academic Editor of PLOS ONE

Thank you for your email and the opportunity to revise our manuscript. We appreciate the time and insights of the reviewers and believe that their comments have helped us to significantly improve the manuscript. We have carefully considered all of the comments and have revised the manuscript accordingly.

Below, we provide a point-by-point response to each of the editor and reviewers' comments.

Journal Requirements:

Comment 1. If the reviewer comments include a recommendation to cite specific previously published works, please review and evaluate these publications to determine whether they are relevant and should be cited. There is no requirement to cite these works unless the editor has indicated otherwise.

Response: Thank you for this clarification.

Comment 2. Please review your reference list to ensure that it is complete and correct. If you have cited papers that have been retracted, please include the rationale for doing so in the manuscript text, or remove these references and replace them with relevant current references. Any changes to the reference list should be mentioned in the rebuttal letter that accompanies your revised manuscript. If you need to cite a retracted article, indicate the article’s retracted status in the References list and also include a citation and full reference for the retraction notice.

Response: Thank you for the detailed instructions. We have reviewed our entire reference list for accuracy, completeness, and correct formatting. We utilized Zotero to specifically check for any retracted articles, and we confirm that none were found in our references.

Reviewer #3:

Thank you for the opportunity to review the revised manuscript. The authors have responded satisfactorily to the initial comments. However, a few inconsistencies as shown below:

Comment 1. In the abstract section, review articles were included while in the method section (i.e., eligibility), reviews were excluded. I suggest you exclude review articles from the scoping review. If they are previous reviews on the subject, you will need to justify why the current review.

Response: We appreciate you pointing out this discrepancy. We apologize for the confusion caused by the error in the abstract. We agree with your suggestion and confirm that review articles are excluded from this scoping review. We have revised the Abstract section to ensure it is fully consistent with the eligibility criteria outlined in the Methods section. (Page 2/Line 28)

Comment 2. Kindly expunge unmarried women from the exclusion criteria. The review focuses on unmarried man. There is no need excluding the unmarried women because by the including only unmarried men, the unmarried women are already excluded.

Response: Thank you for this precise observation. We have removed "unmarried women" from the exclusion criteria section. (Page 5/Line 110)

Comment 3. How many members of the team will be involved in reviewing the selection in Raygan?

Response: Thank you for seeking clarification on this point. We have revised the Data Management and Selection process section to explicitly state that two independent reviewers will be involved in the screening process within Rayyan software. (Page 9/Lines 194-201)

Reviewer #4:

Comment 1. Although the response to my Comment 4 was not entirely satisfactory, I noted that the authors addressed this point in the Key Definitions and Scope section. Overall, the authors have adequately addressed my queries, and I believe the manuscript provides valuable knowledge and is suitable for publication.

Response: We would like to express our sincere gratitude for your time and meticulous review of our manuscript. We are pleased to hear that the revisions made in the "Key Definitions and Scope" section satisfactorily addressed your concerns regarding Comment 4. We truly appreciate your positive evaluation and recommendation for publication.

---

## [Decision Letter · Decision Letter 4]

19 Dec 2025

Thank you for submitting your manuscript to PLOS ONE. After careful consideration, we feel that it has merit but does not fully meet PLOS ONE’s publication criteria as it currently stands. Therefore, we invite you to submit a revised version of the manuscript that addresses the points raised during the review process.

Please submit your revised manuscript by Feb 02 2026 11:59PM. If you will need more time than this to complete your revisions, please reply to this message or contact the journal office at plosone@plos.org . A letter that responds to each point raised by the academic editor and reviewer(s). You should upload this letter as a separate file labeled 'Response to Reviewers'.A marked-up copy of your manuscript that highlights changes made to the original version. You should upload this as a separate file labeled 'Revised Manuscript with Track Changes'.An unmarked version of your revised paper without tracked changes. You should upload this as a separate file labeled 'Manuscript'.

We look forward to receiving your revised manuscript.

Kind regards,

Godwin Banafo Akrong, Ph.D.

Academic Editor

PLOS One

Journal Requirements:

Reviewers' comments:

Reviewer's Responses to Questions

**Comments to the Author**

1. Does the manuscript provide a valid rationale for the proposed study, with clearly identified and justified research questions?

Reviewer #4: Yes

Reviewer #5: Yes

2. Is the protocol technically sound and planned in a manner that will lead to a meaningful outcome and allow testing the stated hypotheses?

Reviewer #4: Yes

Reviewer #5: Yes

3. Is the methodology feasible and described in sufficient detail to allow the work to be replicable?

Reviewer #4: Yes

Reviewer #5: Yes

4. Have the authors described where all data underlying the findings will be made available when the study is complete?

Reviewer #4: No

Reviewer #5: Yes

5. Is the manuscript presented in an intelligible fashion and written in standard English?

Reviewer #4: Yes

Reviewer #5: Yes

You may also provide optional suggestions and comments to authors that they might find helpful in planning their study.

Reviewer #4: Dear Authors,

Thank you for your thorough revisions. You have addressed all my comments clearly and comprehensively. The manuscript is now much stronger, and I have no further concerns. Well done.

Reviewer #5: The manuscript reports a scoping review protocol designed to understand the scope and nature

of evidence related to infertility screening in unmarried men. This is an important issue in

reproductive health and has had limited investigation globally. The protocol is detailed and is

well written, and I believe would lead to a more substantive understanding of the research

question in the international literature.

I have only a few recommendations as follows:

Novelty – the research question is novel and will galvanize future research, policy, and practice

around infertility in unmarried men.

Internal validity – the protocol for the review is well described and if well implemented will

enable replicability and the elicitation of accurate research results. However, the question of

men who may not be married but may have experienced previous fertility or attempts at fertility was not addressed. This should be considered either as an exclusion criterion, or possibly as a limitation of the study.

External validity – I believe the study strengths and limitations as well as the implications of the results for policy and practice should be discussed under the section on discussion.

Grammatical flow and understanding – the manuscript reads well with good flow of ideas.

However, there are a few typos that need to be addressed before the paper is accepted.

**Do you want your identity to be public for this peer review?** For information about this choice, including consent withdrawal, please see our Privacy Policy

Reviewer #4: No

Reviewer #5: **Yes:** Friday Okonofua

---

## [Author Response · Author response to Decision Letter 5]

19 Dec 2025

Dear Dr. Godwin Banafo Akrong, 2025/12/19

Academic Editor of PLOS ONE

Thank you for your email and the opportunity to revise our manuscript. We appreciate the time and insights of the reviewers and believe that their comments have helped us to significantly improve the manuscript. We have carefully considered all of the comments and have revised the manuscript accordingly.

Below, we provide a point-by-point response to each of the editor and reviewers' comments.

Journal Requirements:

Comment 1. If the reviewer comments include a recommendation to cite specific previously published works, please review and evaluate these publications to determine whether they are relevant and should be cited. There is no requirement to cite these works unless the editor has indicated otherwise.

Response: Thank you for this clarification.

Comment 2. Please review your reference list to ensure that it is complete and correct. If you have cited papers that have been retracted, please include the rationale for doing so in the manuscript text, or remove these references and replace them with relevant current references. Any changes to the reference list should be mentioned in the rebuttal letter that accompanies your revised manuscript. If you need to cite a retracted article, indicate the article’s retracted status in the References list and also include a citation and full reference for the retraction notice.

Response: Thank you for the detailed instructions. We have reviewed our entire reference list for accuracy, completeness, and correct formatting. We utilized Zotero to specifically check for any retracted articles, and we confirm that none were found in our references.

Reviewer #4:

Comment 1. Dear Authors,

Thank you for your thorough revisions. You have addressed all my comments clearly and comprehensively. The manuscript is now much stronger, and I have no further concerns. Well done.

Response: We would like to express our sincere gratitude for your review and encouraging comments. We are glad that our revisions addressed your concerns effectively.

Reviewer #5:

The manuscript reports a scoping review protocol designed to understand the scope and nature of evidence related to infertility screening in unmarried men. This is an important issue in reproductive health and has had limited investigation globally. The protocol is detailed and is well written, and I believe would lead to a more substantive understanding of the research question in the international literature.

I have only a few recommendations as follows:

Comment 1. Novelty

– the research question is novel and will galvanize future research, policy, and practice

around infertility in unmarried men.

Response: We would like to thank the reviewer for recognizing the novelty and significance of this research question. We are grateful for your positive evaluation.

Comment 2. Internal validity

– the protocol for the review is well described and if well implemented will enable replicability and the elicitation of accurate research results. However, the question of men who may not be married but may have experienced previous fertility or attempts at fertility was not addressed. This should be considered either as an exclusion criterion, or possibly as a limitation of the study.

Response: Thank you for raising this critical point regarding internal validity. We agree that including men with a history of fertility issues or previous attempts at conception could bias the results, as our focus is on screening. Therefore, based on your recommendation, we have added "Men with a history of previous fertility or attempts at conception" to the exclusion criteria to ensure the study population is clearly defined. (Page 5/Line 110; page 7/Line 153-154)

Comment 3. External validity

– I believe the study strengths and limitations as well as the implications of the results for policy and practice should be discussed under the section on discussion.

Response: We appreciate the reviewer’s recommendation. Accordingly, we have ensured that the Discussion section now covers the study’s strengths, limitations, and the potential implications for policy and practice, as suggested. (Pages 12-13/Lines 262-281)

Comment 4. Grammatical flow and understanding

– the manuscript reads well with good flow of ideas. However, there are a few typos that need to be addressed before the paper is accepted.

Response: Thank you for your positive feedback on the manuscript's flow. We apologize for the typographical errors. We have carefully proofread the entire manuscript again and corrected all identified typos and grammatical slips to ensure the text is error-free.

---

## [Decision Letter · Decision Letter 5]

7 Jan 2026

Infertility Screening in Unmarried Men: A Scoping Review Protocol

PONE-D-25-15832R5

Dear Dr. Mogharabian,

We’re pleased to inform you that your manuscript has been judged scientifically suitable for publication and will be formally accepted for publication once it meets all outstanding technical requirements.

Kind regards,

Godwin Banafo Akrong, Ph.D.

Academic Editor

PLOS One

Additional Editor Comments (optional):

Reviewers' comments:

Reviewer's Responses to Questions

**Comments to the Author**

1. Does the manuscript provide a valid rationale for the proposed study, with clearly identified and justified research questions?

Reviewer #5: Yes

2. Is the protocol technically sound and planned in a manner that will lead to a meaningful outcome and allow testing the stated hypotheses?

Reviewer #5: Yes

3. Is the methodology feasible and described in sufficient detail to allow the work to be replicable?

Reviewer #5: Yes

4. Have the authors described where all data underlying the findings will be made available when the study is complete?

Reviewer #5: Yes

5. Is the manuscript presented in an intelligible fashion and written in standard English?

Reviewer #5: Yes

You may also provide optional suggestions and comments to authors that they might find helpful in planning their study.

Reviewer #5: The manuscript provides a valid rationale for the proposed study, with clearly identified and justified research questions.

**Do you want your identity to be public for this peer review?** For information about this choice, including consent withdrawal, please see our Privacy Policy

Reviewer #5: **Yes:** Prof Friday Okonofua

---

## [Editor Report · Acceptance letter]

PONE-D-25-15832R5

PLOS One

Dear Dr. Mogharabian,

I'm pleased to inform you that your manuscript has been deemed suitable for publication in PLOS One. Congratulations! Your manuscript is now being handed over to our production team.

Kind regards,

on behalf of

Dr. Godwin Banafo Akrong

Academic Editor

PLOS One